

# Design and Implementation of the Detection Software of Wireless Microseismic Acquisition Station Based on Android Platform

Qimao Zhang[1], Shuaiqing Qiao[2], Qisheng Zhang[2], Shiyang Liu[2]

[1] Aerospace Information Research Institute , Chinese Academy of Sciences
[2] Institute of Geophysics and Information Technology (Beijing), China University of Geosciences, China

*Correspondence to*: Qisheng Zhang (zqs@cugb.edu. cn)

**Abstract.** New energy acquisition devices are urgently required to address the increasing global energy consumption and increasing difficulty of energy exploitation. Devices for seismic exploration appear to be small in size, wireless and rapidly becoming more intelligent; hence, a traditional operating platform can no longer satisfy the demand of portable exploration device usage. This study investigates and develops hardware for a wireless microseismic acquisition station, then uses this hardware as a platform to address the distribution of wireless microseismic acquisition stations and deliver monitoring software

based on the Android platform, which is portable, popular and has a large number of users. In large-scale field constructions, software can provide operators with visualised station layouts throughout the process, including positioning, ranging, angle measuring and network monitoring. It also offers a real-time network for monitoring small- and medium-sized microseismic acquisition station arrays under construction as well as other functions, such as intelligent control and real-time data monitoring of the status of the acquisition station. A drainage blast monitoring test is conducted on the system, showing positively

monitored data and accurate results in the inverse operation. Moreover, the software and hardware are proven to be highly stable and portable through a post-construction test, which can help enhance the field construction efficiency.

## 1 Introduction

Non-renewable energies, such as coal, oil and natural gas, provide valuable resources for human survival. Given the rapid development of the industry, the number of mineral resources that can easily be explored has sharply declined, making oil and

gas exploration difficult(Yunlong Liu et al.,2013). This suggests that future exploration may have to look at complex geological areas, such as buried hills and their internal structures, as well as buildings on high and steep mountainsides for oil and gas resources. In that case, not only will the testing depth be increased but the surface and subsurface conditions will also become more complex. All of these abovementioned factors pose new demands and challenges to seismographs(Qisheng Zhang et al.,2016).

Seismic exploration has been valued as an important approach for geophysical exploration. As the main source for seismic exploration, acquisition stations play a central role in resource exploration and energy acquisition(Qiao, S et al.,2018). Microseismic monitoring is a type of seismic monitoring mainly applied to monitoring shocks caused by micromovements,



such as fractures in mines. A microseism is defined as the strain energy released in the form of elastic waves in the stress redistribution process caused by structural break and elongation(Mulargia, F et al.,2016; Kurzon I et al.,2018; Pavlova A et al.,2014). Locating the seismic source with the microseismic data is key to the geometric characteristic analysis of the underground primary and induced fractures, as well as techniques related to effective stimulated reservoir volume (ESRV) and

prediction of trend for future production(Thorne P W et al.,2017). In 1912, Geiger (Germany) proposed a positioning method within the actual domain based on the ideal Earth(Geiger L et al.,1912), which was later applied to the study of microseismic positioning. Anisotropy-based positioning and high-precision positioning based on waveform characteristics have so far been the main methods applied in seismic source positioning(Irina O et al.,2009; Alkhalifah, T et al.,2014; G. A. Meles et al.,2010). As the main source for seismic exploration, the seismic data acquisition stations play a critical role in

resource exploration and energy acquisition(Mazza, S et al.,2012). Similarly, the microseismic data generation also relies on the microseismic data acquisition stations. With the rapid development of science and technology, the evolution of seismic exploration sources tends to focus more on miniaturisation, intelligence and portability whilst maintaining high resolution, high precision, synchronous data acquisition and high-speed data transmission with low power consumption(Qisheng Zhang et al.,2013). From a broader perspective, in the future, seismic data will be collected by a network made up of many miniature

seismic instruments in most areas(Willett K M et al.,2014). At a local scale, data will be acquired from a local area network (LAN) consisting of small seismic instrument arrays in local construction areas. Nevertheless, the current development of seismic exploration sources has seen two bottlenecks in terms of the front-end sensor and the software management platform of a corresponding exploration device.

Computers are used as terminal devices in most of the existing software management platforms. Despite their high computing speed and stable performance, computers are not suitable for many operators in terms of working in the field with a large area and a large distance, considering their poor portability, short life and complicated operations. In addition, most of the current Android software have been applied to seismic data display and rapid earthquake reporting(LIU Jun et al.,2014), whilst only a few have been directly used in the large-scale field construction sites.


Based on the research group has developed the hardware of the acquisition station.This study designed an Android software platform with several distributed wireless microseismic acquisition stations in the laboratory as control targets to solve the abovementioned problems. Compared with traditional computers, Android mobile devices are light and compact with high mobility and excellent computing capability. An Android mobile device can be fitted with 2.4 GHz or 5.8 GHz wireless

transmission units and a Bluetooth module. This setup can help set and control the LAN and ensure the efficiency of wireless transmission. Furthermore, the position information of the Android device user can be quickly inputted to WLAN, thanks to its built-in GPS unit and the GSM/GPRS access module, providing more valid data for construction.





## 2 Hardware Platform

The Android software was designed for the hardware structure of the acquisition system, which monitored and controlled the operation and status of the acquisition system. Designed in the published article "Development of high-precision distributed wirelessmicroseismic acquisition stations", then a brief introduction to its structure.

### 5   2.1 Architecture of the acquisition system

The main tasks completed by the acquisition system included signal synchronisation, data acquisition, AD conversion and data storage. Fig. 1 shows that the vibrations generated from the operation, production and geological movements are firstly received by a geophone. Next, mechanical vibrations were converted into electrical signals, then inputted into the front-end processing circuit for front-end processing. After the analogue signals are processed by primary filtering and impedance

matching in the front-end processing circuit, an amplifier will automatically adjust the amplification factor based on the input signal amplitude. Amplified analogue signals will be further filtered by the filter circuit. Filtered single-ended signals are then converted into differential signals through a differential amplifier, which will then be inputted into an A/D (analogue-to-digital) circuit for conversion. Finally, analogue signals are converted into digital signals, and digital filtering is completed.

The main control unit of the acquisition system consisted of FPGA and CPU (integrated ARM and DSP dual-core processor). FPGA is mainly used for completing multiple tasks that cannot be performed by general-purpose processors: firstly, it is used for allocating and executing any task related to time (e.g. generating the sampling clock, scheduling related strategies and real-time clock); and secondly, it collects the voltage, current, temperature and other information from sensors of the acquisition system and controls the data collection in the acquisition system and data buffering. Finally, it performs wired data

communication, GPS information decoding and time synchronisation. The communication between the CPU core controller and Wi-Fi is achieved through the extended USB of the system. The Android platform also uses that module to communicate with the acquisition system and control the upload and local storage backup (SD card) of data. Additionally, the interaction between the system and the external devices was also completed through core controllers, including LED indicators, on/off switch buttons and battery indications.

### 25   2.2 Construction of the operation network for the acquisition system

A poor outdoor environment together with remote station layouts in large areas can cause data transmitting problems. Accordingly, constructing suitable supported wireless networks is vital in solving the problems and decreasing the difficulty and complexity of collecting field seismic data. Given that most of the construction sites for seismic acquisition are located in field areas with unstable networks, setting up a miniature WLAN in the array is more practical than applying the internet

considering the actual construction. Setting up a miniature wireless WLAN in the array is a critical step. The WLAN should have a high and stable transmission efficiency to minimise the negative influence of topography, and should be easily





maintained and expanded. This study proposed the following WLAN construction plans based on the transmission layer interfaces provided by the existing distributed wireless microseismic acquisition stations as well as actual applications:

 The 2.4 GHz WLAN to provide access points and the 5.8 GHz WLAN for remote data transmission were adopted in this plan . A single acquisition station was connected to a single 2.4 GHz omni-directional AP through the TCP/IP protocol by binding

5 SSID, connecting the MAC address of the device and limiting the number of connected devices. The data were transmitted from the network cable to the 5.8 GHz directional antenna, then to the hub station AP through directional transmission. Whilst the 2.4 GHz wireless signal was largely affected by atmospheric attenuation, the 5.8 GHz WLAN proved to have a higher data transmission rate and a greater signal radiation distance under the same transmitting power of the directional antenna. The WLAN constructed in this manner had a clear network topology, and can easily be maintained.

## 3 Design and Implementation of Monitoring Software for the Android Host Computer

### 3.1 Android platform

Android is a free and open-source Linux-based operating system mainly used for mobile devices, such as smartphones and tablets. The Android system architecture has employed a layered architecture composed of four layers. The layers from top to bottom are the application layer, Android framework, Android runtime and Linux kernel. Android 9.0 is the latest version.

### 3.2 Software demand analysis

(1) The software should have visualised layouts (i.e. custom maps) to display the specific location of the arranged wireless microseismic acquisition stations on the map based on the field construction requirements. In addition, offline maps are required to ensure the normal use of the software when network connection is unavailable outdoors.

(2) During the station distribution, the relative distance and the angle between the acquisition and host stations as well as the communication status of the WLAN must be checked.

(3) The parameters and status of all acquisition stations in the current network should be configured and controlled through an Android mobile device upon the completion of the station distribution.

(4) After the acquisition stations start collecting data, the software should be able to monitor the status of the acquisition station, including the temperature, power, remaining storage space and the number of satellites connected to the GPS to help operators monitor the acquisition stations.

(5) The software should achieve real-time data collection from the acquisition stations, which can be displayed in multi-channel waveforms to observe the real-time acquisition.

### 3.3 Software architecture design

According to the demand analyses in Section 3.2, the Android software should be equipped with WLAN access and communication, control, monitoring, real-time position, waveform display, data storage and other functions of the multiple



distributed wireless microseismic acquisition stations in the WLAN. Additionally, it should be utilised to obtain the location and the network status of the Android mobile device where the software is installed. Fig.2 shows the designed software architecture interface.

The software design follows the MVP software design pattern, where MVP stands for model, view and presenter. The MVP has evolved from the MVC pattern, which is short for model, view and controller. The presentation logic and the business logic are separated in the MVP to reduce coupling. In a traditional MVP architecture, the model layer is used for business logic and solid modelling. The view layer is an interface displaying an updated view to the user and receiving the data inputted by the user. The presenter layer is used to separate the model and view layers and reference through the interfaces of the view and

the model. It also loads data and updates UI. Business logic is performed by the interfaces in the presenter layer. In the traditional pattern, the system is packed with implementation classes and interfaces when processing simple requests. The view layer is rarely modified in the actual project. Many bugs of business logic must be fixed during programme upgrade and iteration. Therefore, in the MVP officially defined by ⓒGoogle, logic processing has been added to the presenter layer such that the model layer has less functions and only provides data models. In the MVP pattern, the implementation classes of the

view layer, such as activity and fragment, are responsible for processing the life cycle of the presenter layer, avoiding the storage leakage of activity. Activity does not process specific business logic. The interfaces in the presenter make it easier to conduct unit tests. The MVP pattern can generally offer a lower coupling in programming, a clearer software structure and improvement in code flexibility and maintainability.

A framework with a single activity and multiple fragments in the MVP pattern was adopted. Whilst the framework has a clear logic with one activity on one page, the activity may put pressure on the loading of the mobile device. The system is also more likely to slow down and freeze because of frequent jumps, resulting in a poor immersive experience for users. On the contrary, the framework with a single activity and multiple fragments can reduce the amount of codes whilst ensuring a clear logic to decrease the difficulty in interacting among different interfaces. Moreover, the transition of pages when swiping smoothly can

enhance the fluency of the software and immersive experience for users.

### 3.4 Software design process

Fig. 3 shows the main process designed for the Android software. After connecting into the established WLAN, the connection and the communication of the Android mobile device will be conducted by accessing the static IP addresses of a single or multiple acquisition station(s) in the WLAN. Station-searching broadcast packets will be sent to the broadcast address in the

WLAN through the UDP protocol in the software. Alternatively, the packets can be sent to a series of consecutive static IP addresses in loops. In the defined protocol, the acquisition station(s) will send addresses to the broadcast packets to respond to the status packets and obtain the position and status of all acquisition stations in the WLAN. Finally, real-time data interaction and work status control will be completed by establishing the TCP/IP connections with all the acquisition stations.





## 4 Development and Implementation of the Android Software

### 4.1 Development environment for the Android programme

The Android software was written in Java with Android Studio. The API level for the Android SDK packages was 26, and the version strings in JDK 8 were 1.8.0. Meanwhile, BlackBerry Priv (Android 6.0) and Samsung S3 (Android 7.1) with a 3.0.1

base kernel were used for testing and running the software.

### 4.2 Key technologies in Android modules

(1) Schemes for the communication between the Android mobile device and the acquisition stations: TCP/IP is a connection-oriented protocol focusing on point-to-point communication. Data communication can only be established based on a reliable connection (i.e. three 'handshakes') between the client and the server. Contrary to TCP/IP, UDP is a protocol that

uses connectionless communication. Although the UDP can enjoy a high-speed interaction without building connections, its reliability is lower than that of TCP/IP. UDP communication should be adopted to quickly receive reply packets through scanning all acquisition stations in WLAN when a large number of acquisition stations assess WLAN. Considering that a long time may be required for retrieving a great deal of data in real-time data monitoring, TCP/IP should be adopted to ensure stable connection and high reliability. The above two types of communication in the Android platform can be

delivered by using Socket and DatagramSocket classes to specify the static address of the acquisition stations and the broadcast address of the WLAN.

(2) Communication protocol and unpacking technology for the Android software and acquisition stations: firstly, packet analysis is difficult in communication transmission under the existing protocol when the software is written by Java. For instance, Java does not support the structure type in C, and it also has transmission problems, such as big- and little-endian

formats. To solve this problem, JavaStruct, which is a third-party open source packet, should be introduced to convert the structure and the class through the implementation classes in the open source packet. At the same time, byte array shifting can tackle big and little endian in transmission. Secondly, different value ranges might be caused by different language environments and data types. For example, unsigned types like uint and uchar in C are not supported in Java. Moreover, the same data type may occupy different numbers of bytes. For example, char occupies 1 byte in C, but 2 bytes in Java.

We solve these problems by replacing unsigned data types with high-precision data types and data types with the same byte or shifting byte array in the process of converting the structure and the class.

(3) Key positioning technologies in the station layouts: an Android mobile device has GPS positioning; hence, the SDK of the Baidu Map in BaiduLBS, which is a third-party open platform, can be introduced to achieve custom maps, navigation and other functions. SDK provides an interface for developers to access the Baidu Map database. If positioning is allowed,

Baidu's satellite, traffic conditions and 3D maps can be used by calling the interface. The Baidu Map is based on the overlapping of multiple map layers to display different functions. A marking layer is added on top of the source map. The latitude and longitude coordinates of the accessed WLAN can be obtained through the reply packet returned by UDP



communication. Layer marks are also added on the marking layer. The status data of the relevant acquisition stations are included in the pop-up box to mark the increase of the click events.

(4) Real-time data imaging technology: this software uses MPAndroidChart, which is a third-party open source charting library, to receive real-time imaging data. MPAndroidChart can display different types of images, such as polylines, scatter and

histograms, with X-/Y-axis scaling and a customised outlook. The X- and Y-axis data sets will be generated once the real-time data are received from the blocking array. They will then be transferred to the line chart control unit in the MPAndroidChart to achieve data imaging in real time.

## 4.3 Specific functions of the Android software

The software can be installed on mobile phones with an Android 4.0 or above OS and can self-adapt to the screen size.

WIFIFragment will be loaded first and displayed on the first page. The main functions include online/offline custom maps, layouts, control, status monitoring, data imaging and parameter settings of the acquisition stations.

(1) Station-searching function in WLAN: all acquisition stations of WLAN (SSID: CUGB_Microseism_2.4G) currently accessed can be scanned with the scan button in the software. The station number and the working status of all acquisition stations are displayed with a list in the blank space. TCP/IP can be connected by inputting the statistic IP of the acquisition

station.

(2) Custom maps: a custom map can be accessed through the map button on the Wi-Fi page. Information, such as the coordinates of all acquisition stations in WLAN, can be obtained through UDP and displayed in the map. The angle and the distance between the acquisition and base stations can be measured after inputting the coordinates of the base station (Fig. 4). In addition, break-point resume downloading of offline maps and positioning of the Android mobile devices are

provided in the function of the custom map.

(3) Control of the acquisition stations: an acquisition station can be controlled to start or stop data acquisition after a stable TCP/IP connection is established with the target acquisition station.

(4) Setting of the acquisition parameters: each channel gain of the target acquisition station must be set. The acquisition station will be controlled to start collecting data after setting the relevant parameters. The set parameters and command packet

will be sent to the acquisition stations. Assuming that the acquisition parameters are set after the start of the data acquisition, the setting will be dispatched before the next start of data acquisition.

(5) Real-time data imaging: the software can analyse the multi-channel data returned by real-time transmission from the target acquisition station and visualise the data in waveforms (Fig. 5). The channel can be selected on the waveform page.

(6) Real-time status monitoring of the acquisition station: the software can monitor the status of all the acquisition stations

connected to WLAN. The monitoring data included the current working status, working temperature, remaining space of the SD card, battery power and number of satellites connected to the GPS, among others. The monitoring function can also be displayed on the custom map. The status information of the acquisition station can be displayed by tabbing the mark of each acquisition station on the map.

## 5 Monitoring Test of Drainage and Blasting

This test aimed to measure the blasting location and the range affected by the blasting in the previously designed acquisition stations. Sixteen microseismic acquisition systems were used in the test to conduct real-time data acquisition, data recording and post-monitoring analysis. The acquisition system was a 24-bit AD acquisition system with a sampling rate of 1 kHz and a
dynamic range of 137 dB.

According to the data collected by the collection station, the processor is used to process and detect the collected data, and the picked up vibration events are counted in a histogram. The horizontal axis of the microseismic time-frequency diagram is the monitoring time and the vertical axis is The number of micro earthquakes per unit time. As can be seen from the above figure,
in the time course, the microseismic distribution can be divided into three clusters. The frequency of microseismic rupture is low, indicating that there is a rupture process; the frequency of microseismic events during the process of fracture extension, sanding and shut-in Lower, fracturing construction is smooth.

Figs. 6 and 7 respectively introduce the time-frequency and space-frequency relationship diagrams of the number of seismic
events triggered. According to the number of fracturing vibration events collected by multiple microseismic acquisition stations and the difference in time difference between each event, it can be Reverse the location of fracturing and the trend of fracturing fractures. Fig. 8 is the geometric parameters of the crack displayed by the host computer.

According to the data processing and calculation of the central station, the geophysical data can be used to invert the original
fracture trend map and three-dimensional fracture map of underground fractures (as shown in Fig. 9).

According to the experimental data analysis, the acquisition system can complete the exploration task very well under the Android software monitoring. The acquisition system can properly collect data and run stably. The positions of the seismic sources and the induced seisms were obtained through the experimental data inversion. Ultimately, the degree and the trend of
the fracture fission were calculated.

## 6 Conclusion

The distribution of stations and the status of data monitoring in the large-scale construction of field seismic data collection were studied herein. A microseismic acquisition system was designed for application to complex outdoor environments. Based on that, this study introduced software for the distribution and monitoring of distributed wireless microseismic acquisition
stations based on the Android platform. The software combined UDP and TCP/IP to achieve data communication and monitoring with the acquisition stations. This study also provided solutions to the incompatibility between different programming languages in the current communication protocol. Visualised station layout and real-time data imaging were



accomplished using third-party libraries. All construction processes, including station layouts, data acquisition and monitoring of the acquisition stations, were precisely designed. After many outdoor experiments, the software was proven to be easy to use with a simple operation. It had a stable performance and exhibited smooth runs. The software can make the operators' lives easier and can also reduce the workload of data observers at the host station. Construction workers can collect construction

data in any area covered with WLAN by using only an Android mobile device, rather than monitoring the status of the acquisition station at a fixed position with a computer/laptop. In this way, the outdoor workload of an operator can be significantly reduced. These results indicate that the software has a great value for future applications and marketing promotions. The outdoor tests obtained positive data and accurate test results, as expected. This study has laid a foundation for the implementation of the acquisition system in future mining explorations.

**Acknowledgment**

This work was supported by the Natural Science Foundation of China (No.41574131& No.42074155), the National Key Research and Development Program of China (No.2017YFF0105704), the National "863" Program of China (No. 2012AA06110203), PetroChina Innovation Foundation (No. 2019D-5007-0302),and  the Fundamental Research Funds for the

Central Universities of China.

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





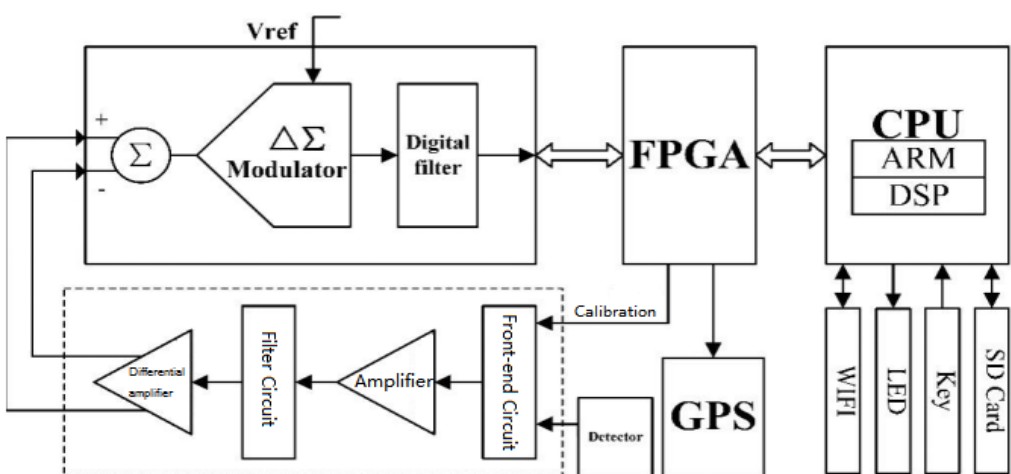

**Figure 1: Overall Architecture of the Acquisition System**

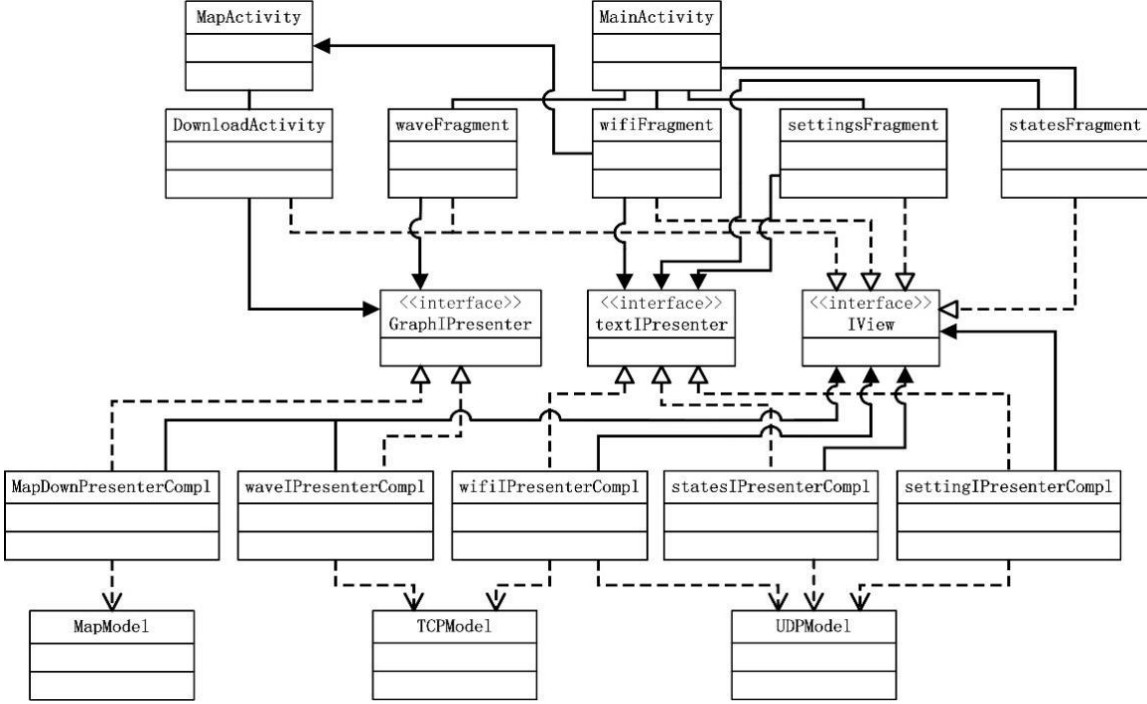

**Figure 2: Application Programming Interface**



**Figure 3: Main Process of the Software**



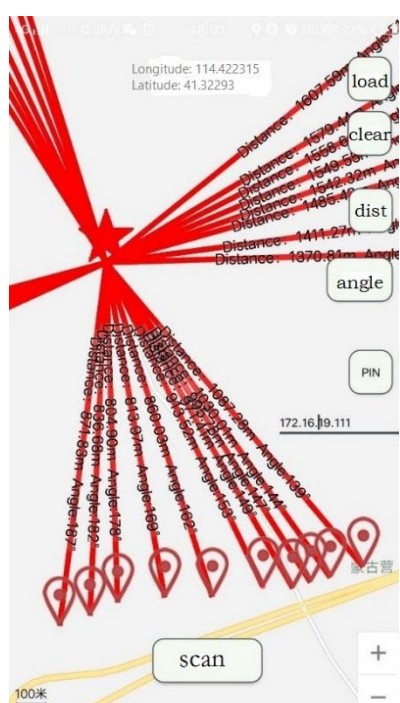

**Figure 4: Custom map,positioning, Angle measurement and Ranging(ⒸBaidu Maps2019)**

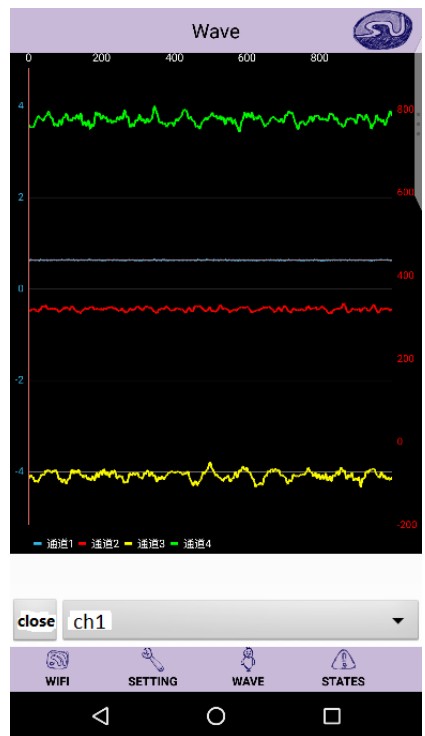

**Figure 5: Three-channel waveform**




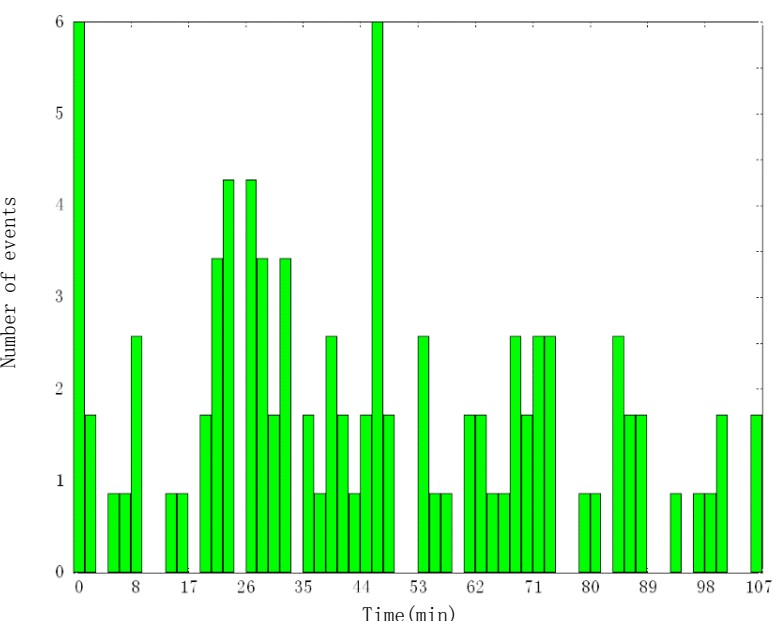

**Figure6: Earthquake event time-frequency diagram**

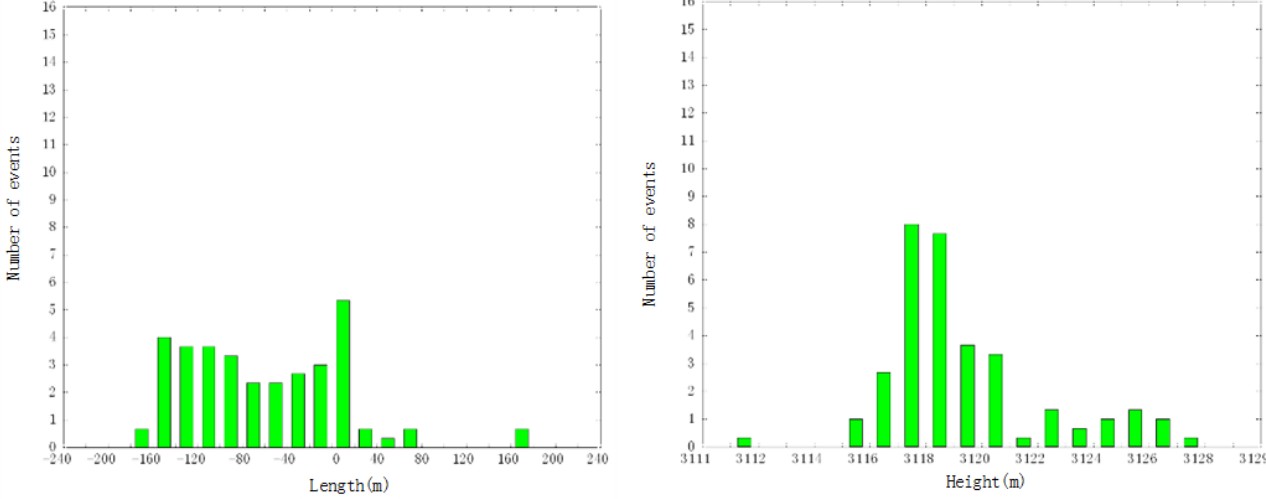

**Figure 7: Frequency chart of events in different dimensions (left: horizontal; right: vertical)**



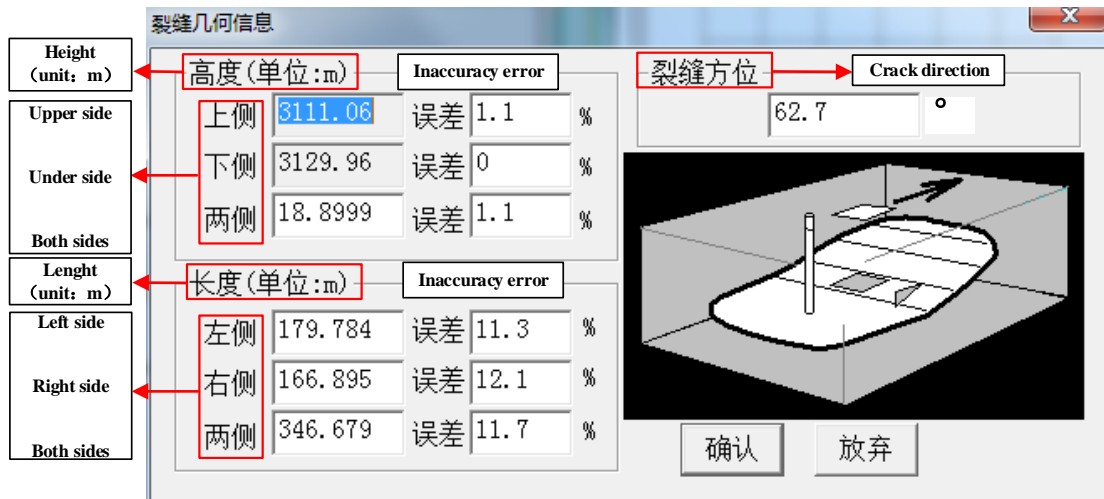

**Figure 8: The upper computer monitors the geometric information of fracturing fractures**

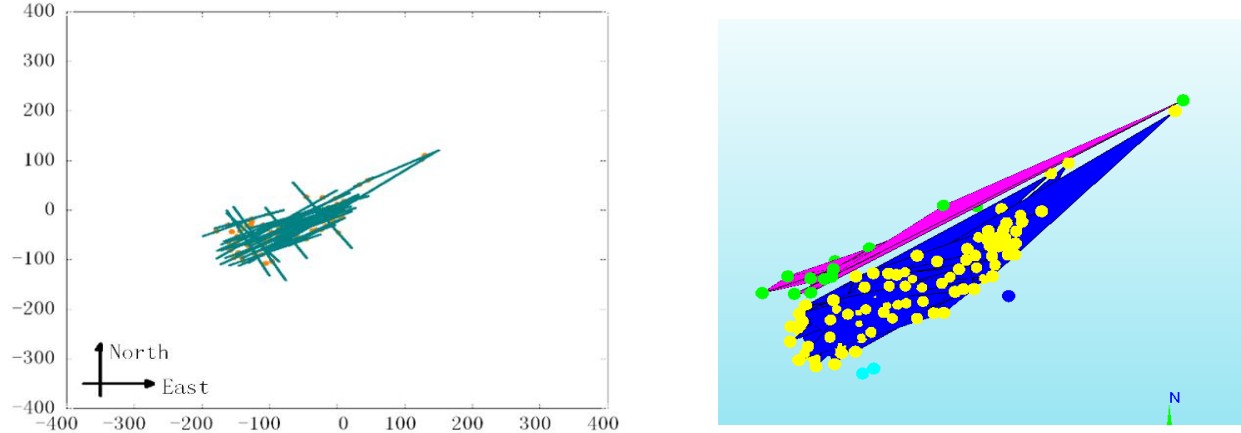

**Figure 9: Three-dimensional view of cracks (left: original fracture strike structure; right: three-dimensional view)**

