# Peer review of "Design and Implementation of the Detection Software of Wireless Microseismic Acquisition Station Based on Android Platform"

_Geoscientific Instrumentation, Methods and Data Systems, 2020_

## Referee Comment (RC1) · Anonymous Referee #1 · 9 Nov 2020

The idea of this manuscript is original and unique, and it describes the technical realization of the wireless microseismic acquisition station for geophysical exploration. I have attached below several comments arising from the manuscript, which might be helpful to further improve the quality of the publication of the paper.

1.The format of the article references is not uniform, and the references need to be enriched; 2.The experimental part of the paper mentions the performance of the collection station and the dynamic range of the sampling rate, but the key parameter of SNR is not involved and needed to be added; 3.Figure 8 in the text is a little bit messy, and should be adjusted appropriately.

[Figure]

In terms of geophysical detection instruments, it has solved the difficulty of deploying and patrolling stations in the field. It is strongly recommended to make minor modifications to be accepted.

Please also note the supplement to this comment:
https://gi.copernicus.org/preprints/gi-2020-36/gi-2020-36-RC1-supplement.pdf
* * *

---

## Author Comment (AC1) · 10 Nov 2020

Dear editor Thank you for your comments on my manuscript during your busy schedule. Reply to your question as follows: 1. Some references have been added for your comments; 2. Added the accuracy index of the instrument in the test part of Chapter 4 of the article; 3. Modified to Figure 8. Thank you.

---

## Referee Comment (RC2) · Anonymous Referee #2 · 12 Dec 2020

1. The article mentioned that the mobile terminal communicates with the collection station through wireless means, how far the communication distance between the mobile device and the collection station can beïïj§ 2. Figure 4 shows the positioning position and angle of the mobile device test collection station. In actual measurement, how many meters can the positioning accuracy of the collection station reachïïj§ 3. The abscissa and ordinate of Figure 9 do not indicate the unit of measurement and need to be supplemented. 4. In addition to being used on the Android platform of mobile phones, can the software system be implanted in other mobile devices?

The article has a rigorous overall structure and innovative ideas. It provides more

convenient station inspection and control functions in geophysical exploration. It is recommended to use it after modification.
* * *

---

## Author Comment (AC2) · 13 Dec 2020

1. The networking mode of the collection station is "collection station-bridge-central station". In this mode, the communication distance between the collection station and the mobile device can reach 2km; 2. The equipment adopts GPS positioning system. During the on-site deployment of the collection station, the measured positioning accuracy is 2.5mCEP; 3. Thank you for your suggestions, the corresponding parts have been modified according to your suggestions; 4. After obtaining the root authority of the corresponding system, it can be transplanted.

[Figure]

https://doi.org/10.5194/gi-2020-36, 2020.

---

## Author Response (AR1)

**Reviewer #1**

1,The format of the article references is not uniform, and the references need to be enriched;

**Answer**:After inspection, there are indeed some format problems, which have been corrected according to your comments;

2,The experimental part of the article mentions part of the performance of the collection station and the dynamic range of the sampling rate, but the key parameter SNR is not involved and needs to be added;

**Answer**:Indicators have been added to the conclusion of the article;

3,Figure 8 in the text is a bit messy, adjust appropriately. The overall structure of this film is clear and innovative.

**Answer**:The content in the picture has been modified according to your suggestions.

**Reviewer #2**

1,The article mentions how far the communication distance between the devices can reach through wireless communication between mobile terminals;

**Answer**:The networking mode of the collection station is "collection station-bridge-central station". In this mode, the communication distance between the collection station and the mobile device can reach 2km;

2,The signal-to-noise ratio of the acquisition equipment is 137dB, under what test conditions;

**Answer**:The equipment adopts GPS positioning system. During the on-site deployment of the collection station, the measured positioning accuracy is 2.5mCEP;

3,The signal-to-noise ratio of the acquisition equipment is 137dB, under what test conditions;

**Answer**:Thank you for your suggestions, the corresponding parts have been

modified according to your suggestions; suggestions.

4,When the mobile system locates the acquisition equipment, how many meters can the accuracy be?

**Answer**:Can be transplanted.

Thank you for reviewing the manuscript in your busy schedule.